# Is the Gridded Data Accurate? Evaluation of Precipitation and Historical Wet and Dry Periods from ERA5 Data for Canadian Prairies

Thiago Frank [1,*], Carlos Antonio da Silva Junior [2], Krystopher J. Chutko [1], Paulo Eduardo Teodoro [3], José Francisco de Oliveira-Júnior [4] and Xulin Guo [1]

1   Department of Geography and Planning, University of Saskatchewan, Saskatoon, SK S7N 5C8, Canada
2   Department of Geography, State University of Mato Grosso (UNEMAT), Sinop 78555-000, Brazil
3   Department of Agronomy, Federal University of Mato Grosso do Sul (UFMS), Chapadão do Sul 79560-000, Brazil
4   Institute of Atmospheric Sciences, Federal University of Alagoas (UFAL), Maceió 57072-970, Brazil
*   Correspondence: thiago.frank@usask.ca

**Abstract:** Precipitation is crucial for the hydrological cycle and is directly related to many ecological processes. Historically, measurements of precipitation totals were made at weather stations, but spatial and temporal coverage suffered due to the lack of a robust network of weather stations and temporal gaps in observations. Several products have been proposed to identify the location of the occurrence of precipitation and measure its intensity from different types of estimates, based on alternative data sources, that have global (or quasi-global) coverage with long historical time series. However, there are concerns about the accuracy of these estimates. The objective of this study is to evaluate the accuracy of the ERA5 product for two ecoregions of the Canadian Prairies through comparison with monthly means measured from 1981–2019 at ten weather stations (in-situ), as well as to assess the intraseasonal variability of precipitation and identify dry and wet periods based on the annual Standardized Precipitation Index (SPI) derived from ERA5. A significant relationship between in-situ data and ERA5 data (with the $R^2$ varying between 0.42 and 0.76 ($p < 0.01$)) was observed in nine of the ten weather stations analyzed, with lower RMSE in the Mixed Ecoregion. The Mean Absolute Percentage Error (MAPE) results showed greater agreement between the datasets in May (average R value of 0.84 and an average MAPE value of 32.33%), while greater divergences were observed in February (average R value of 0.57 and an average MAPE value of 50.40%). The analysis of wet and dry periods, based on the SPI derived from ERA5, and the comparison with events associated with the El Niño-Southern Oscillation (ENSO), showed that from the ERA5 data and the derivation of the SPI it is possible to identify anomalies in temporal series with consistent patterns that can be associated with historical events that have been highlighted in the literature. Therefore, our results show that ERA5 data has potential to be an alternative for estimating precipitation in regions with few in-situ stations or with gaps in the time series in the Canadian Prairies, especially at the beginning of the growing season.

**Keywords:** precipitation; gridded data; observed data; hydrological cycle; ENSO

## 1. Introduction

Precipitation is a key input to the hydrological cycle and therefore directly affects all ecological processes that occur on Earth's surface. Given its importance, knowing the location and intensity of precipitation is essential information in the spatio-temporal assessment of this phenomenon [1–4]. Historically, precipitation is measured at surface weather stations (in-situ), but in countries with a large territorial extension, such as Canada, where the network of in-situ stations usually is not evenly distributed in the territory and the historical information varies considerably, relying solely on in-situ data can be

challenging [5–7]. It should also be considered that the implementation and maintenance of weather stations means a substantial financial investment, which may be unfeasible in some locations. To overcome such limitations, there are precipitation estimates that have global (or quasi-global) coverage with long historical time series [8–10]. These estimates are based on satellites that observe Earth's surface, from algorithms that analyze the vertical profile of the atmosphere [11,12] followed by the combination of satellite information with information from in-situ stations [13], or through the interpolation and extrapolation of historical data from different locations that establish spatial and temporal patterns. The main limitation of the estimates is the reliability of the results since discrepancies between estimated and measured data have already been observed [14]. It is also worth noting that the discrepancies between estimated and observed data tend to be greater at high latitudes, where low-intensity precipitation events (both for rain and snow) are more difficult to be detected by sensors [15]. Finally, passive sensors may show greater deviations when estimating precipitation in cold areas due to the characteristics of the ice that covers the land for long periods of time [12].

By providing historical data systematically, precipitation products benefit several climatological, hydrological, and environmental studies [8–10]. In the literature, various precipitation products were used for statistical validation [16–18], to identify extreme weather events [19–21], and to evaluate climatic variability [22,23]. It is worth mentioning that each precipitation product has specific characteristics, such as the coverage area, time frame, and spatial resolution [8,9], as well as the use of several different parameterization schemes [24]. Currently, there are several sources of precipitation data that offer recent and historical gridded information at a range of spatial scales. Global Precipitation Measurement (GPM) is one of the most widely used products, has quasi-global scale, and uses hourly data collected by a set of satellites with a spatial resolution of approximately 10 km [12]. GPM is a continuation of the Tropical Rainfall Measuring Mission (TRMM), which operated between 1997 and 2015 on an quasi-global scale with a spatial resolution of approximately 25 km [11]. There are also products that combine satellite information, precipitation estimates, and information from in-situ stations, such as the Climate Hazards Group of University of California (CHIRPS), which has a spatial resolution of approximately 5 km [13,21]. Finally, there are products that are obtained from the data interpolation, such as Daymet, which covers North America (NA). Therefore, there are different methods for estimating current and historical precipitation in different locations around the globe in a simplified way for the end user, with a standardized approach, and without the absence of data.

Unfortunately, in certain regions of Earth the availability of precipitation data is limited. Estimates from GPM, TRMM, and CHIRPS do not cover high latitudes and exclude almost the entire territory of countries located in high latitudes, such as Canada. Despite covering all Canadian territory, Daymet's data is restricted to NA, a fact that makes it impossible to replicate the same methodology in other parts of the world. In this context of data limitation, the ECMWF Reanalysis 5th Generation (ERA5), developed by the European Center for Medium Range Weather Forecasts [25] and published by Copernicus Climate Change Service, appears as an interesting dataset. This is because precipitation re-analysis data with spatial resolution of approximately 9 km since 1979 has been globally available [26]. Unlike other reanalysis products, ERA5 has an important remote sensing component. ERA5 uses as input several observations such as radiance, ozone, wind, temperature, soil moisture, and humidity, which are collected by geostationary satellites, polar orbiting satellites, ground-based radargauge, radiosondes, dropsonde, and others [27]. According to [27] the number of daily observations in ERA5 increased 32 times (from 0.75 million to 24 million) in a period of 40 years, with the radiation observations made by satellite being largely responsible for the expressive increase in observations between 1979 and 2019, which strongly contributes to the improvement of ERA5 model and its results compared to its predecessors. Despite not being a primary source of data, ERA5 can be understood as a hub of climate information from several sources that has models to estimate precipitation

and other climate variables in a systematic way on a global scale and available to the public free of charge. However, there are still divergences related to the accuracy in cases of extreme precipitation. As is the case with [28], in which the authors concluded that ERA5 underestimates extreme events, while [29] observed several similarities between ERA5 and historical data from in-situ weather stations.

A potential derivation of the ERA5 data is the calculation of the Standardized Precipitation Index (SPI) [30], an index widely used for monitoring dry and wet periods, and for the evaluation of severe droughts [21]. Studies like [31,32] derived the SPI from CHIRPS time series to detect the occurrence of climatic extremes. This type of study is still rare in Canada, especially in the Prairies region, an arid but extremely important region for the country's agricultural production [33]. We found SPI-based studies applied to Canadian Prairies like [34,35], both in a national scale but without the use gridded data. To our knowledge, no other studies were found that derived the SPI based on ERA5 data for the Canadian Prairies to identify dry and wet periods.

Both the Mixed Ecoregion and the Moist-Mixed Ecoregion are regions located in the Prairie Provinces of Alberta and Saskatchewan, where livestock and grain production are fundamental to the local economy and as a source of food for the domestic and foreign markets [36–38]. Therefore, both ecoregions are highly dependent on precipitation and are considered regions susceptible to drought [36]. In this context, the objectives of this study are: (i) to verify the accuracy of the ERA5 product for the Mixed Ecoregion and for the Moist-Mixed Ecoregion; (ii) to assess the intraseasonal variability of precipitation from the ERA5 data; and (iii) identify dry periods and wet periods based on the annual SPI derived from ERA5.

## 2. Materials and Methods

### 2.1. Study Area

The study area comprises the Mixed Ecoregion and the Moist-Mixed Ecoregion, located in the Prairie Provinces of Alberta (AB) and Saskatchewan (SK). The two ecoregions have a total area of 233,595 km$^2$ (Figure 1), with a predominant dry and cold climate [36]. In the Mixed Ecoregion, the average winter and summer temperatures are $-10$ °C and $16$ °C, respectively, with an annual precipitation of approximately 300 mm [39]. The Moist-Mixed Ecoregion tends to be less susceptible to long periods of drought than the Mixed Ecoregion, with average winter and summer temperatures of $-11$ °C and $15.5$ °C, respectively, with an annual precipitation of 400 mm [36].

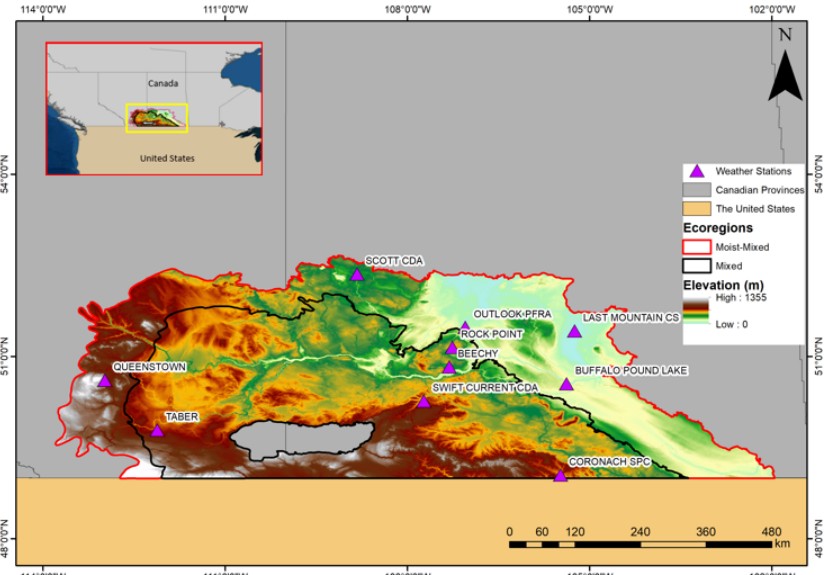

**Figure 1.** Study area, including the Mixed and the Moist-Mixed Ecoregions, and the weather stations and elevation (m)—(USGS/SRTMGL1_003 [40]).

### 2.2. Organization and Treatment of the Precipitation Time Series

Daily precipitation data was acquired from the Canadian government website (https://climate.weather.gc.ca/historica_data (accessed on 1 March 2021)) which provides historical weather information from more than 8800 weather stations across the country (most with short or sporadic temporal coverage). In a first verification, 646 meteorological stations were found in the study area. However, after filtering the consistency of the historical data for each station, we found only ten stations with a sequence of data without gaps from 1981 to 2019. Therefore, for the present study, daily precipitation information from ten different in-situ stations located in the study area was used (Table 1).

**Table 1.** Weather stations used to validate ERA5 data with Station ID, Latitude (°), Longitude (°), Elevation (m), and Ecoregion.

| Station Name | Province | Station ID | Latitude (°) | Longitude (°) | Elevation (m) | Ecoregion |
|---|---|---|---|---|---|---|
| Taber | AB | 2315 | 49.79 | −112.12 | 811 | Mixed |
| Beechy | SK | 3071 | 50.83 | −107.31 | 660 | Mixed |
| Rock Point | SK | 3142 | 51.15 | −107.26 | 725 | Mixed |
| Swift Current CDA | SK | 3157 | 50.27 | −107.73 | 825 | Mixed |
| Coronach SPC | SK | 3172 | 49.05 | −105.48 | 756 | Mixed |
| Queenstown | AB | 2295 | 50.61 | −112.98 | 940 | Moist-Mixed |
| Buffalo Pound Lake | SK | 2859 | 50.55 | −105.38 | 588 | Moist-Mixed |
| Last Mountain CS | SK | 2942 | 51.42 | −105.25 | 497 | Moist-Mixed |
| Scott CDA | SK | 3259 | 52.36 | −108.83 | 660 | Moist-Mixed |
| Outlook PFRA | SK | 3318 | 51.48 | −107.05 | 541 | Moist-Mixed |

### 2.3. ERA5 Data

ERA5 is a global atmospheric reanalysis product developed by ECMWF using 4D-Var data assimilation techniques in the 41r2 cycle [25]. In this study, the recent product launched by ECMWF designated as ERA5 was used [24]. ERA5 has important changes compared to ERA-Interim, including improved numerical models, such as the Cycle 41r2, and data assimilation schemes, greater spatial and temporal resolution, assimilation of more observations, and improved versions of observations from forcing data sets [41]. The longer time series is extremely interesting for characterization studies of the climate of a region and one of the biggest benefits over remote sensing products that have a shorter observation history. ERA5 estimates for rain and snow are based on a wide range of satellite sensor and ground-based observational inputs including column water vapor, relative humidity, cloud liquid water, and precipitation [27]. These data can be extremely useful for obtaining more accurate results in relation to measurements made only by satellites, as verified by [42] in temperate regions of China, a region that presents similar characteristics to the area of interest of this study.

For this study, the ERA5 pixels corresponding to the pairs of coordinates of each of the weather stations, covering January 1981 through December 2019, were downloaded from the JavaScript programming interface of the Google Earth Engine platform [43] via the product "ee.ImageCollection" ("ECMWF/ERA5_LAND/MONTHLY"). Subsequently, the monthly amounts were added for all the years evaluated and then the 39-year precipitation data was reduced to the monthly average.

### 2.4. Standardized Precipitation Index (SPI)

To determine the SPI [30], the Gamma distribution was used and is defined in Equation (1):

$$f(x) = \frac{1}{\Gamma(a)\beta^a} x^{a-1} e^{-\frac{x}{\beta}} \tag{1}$$

where: $a > 0$ ($a$) shape parameter (dimensionless); $\beta > 0$ ($\beta$) scale parameter (mm); $x > 0$ ($x$) total precipitation (mm); and $\Gamma$ ($\alpha$) Gamma function $= \Gamma(a) = \int_0^\infty x^{a-1} e^{-x} dx$.

All parameters and the gamma function were adjusted for the cumulative frequency distribution for precipitation based on ECMWF Reanalysis v5 (ERA5) data on the annual scale. We calculated the $\alpha$ and $\beta$ parameters of gamma for each pixel referring to the location of the weather stations on the annual scale. The maximum-likelihood method was used for estimating the $\alpha$ and $\beta$ [44,45]. Calculations of parameters $\alpha$ and $\beta$ were performed to find the cumulative probability of an observed precipitation event for the adopted scale. The cumulative probability is given by Equation (2).

$$F(x) = \int_0^x f(x)dx = \frac{1}{\Gamma(a)\beta^a} \int_0^x x^{a-1} e^{-\frac{x}{\beta}} dx \tag{2}$$

The annual SPI values were classified into wet and dry periods, as used by [30], according to Table 2.

**Table 2.** General SPI classification [30].

| SPI Values | Classification |
|---|---|
| >2.0 | Extreme Wet |
| 1.5 to 1.99 | Severe Wet |
| 1.0 to 1.49 | Moderate Wet |
| 0.99 to −0.99 | Normal |
| −1.0 to −1.49 | Moderate Drought |
| −1.5 to −1.99 | Severe Drought |
| <=−2.0 | Extreme Drought |

*2.5. Statistical Metrics*

To compare the precipitation data from weather stations and ERA5, the coefficient of determination ($R^2$) was used to determine the level of data correlation, with greater $R^2$ values indicating greater similarities between the datasets, the Root Mean Square Error (RMSE, mm) was used to assess precision and accuracy, with lower RMSE values indicating greater similarities between the datasets., and the Mean Bias Error (MBE, mm) was used to identify the average bias in the prediction, with lower MBE values indicating greater similarities between the datasets. Finally, the Mean Absolute Percentage Error (MAPE), which is a time series analysis approach using the absolute Euclidean distance between paired series, was used to identify at which station and when the greatest differences and greatest similarities occurred, with lower MAPE values indicating greater similarities between the datasets. The equations are listed below:

$$R^2 = 1 - \frac{\sum_{i=1}^n \left(\left|E_i - \overline{O_i}\right|\right)^2}{\sum_{i=1}^n \left(\left|O_i - \overline{O_i}\right|\right)} \tag{3}$$

$$RMSE = \sqrt{\frac{\sum_{i=1}^n (O_i - E_i)^2}{n}} \tag{4}$$

where $n$ = number of observations; $O_i$ = $i$-th value of observed data and $E_i$ = $i$-th value of estimated data; and $\overline{O_i}$ = mean of the observed data.

$$MBE = 1/n \sum_{i=1}^n (P_i - O_i) \tag{5}$$

where $O_i$ is the observation value and $Pi$ is the forecast value.

$$MAPE = \frac{100}{n} * \sum_{i=1}^n \frac{|yi - \overline{y_i}|}{yi} \tag{6}$$

where $yi$ is the actual observations time series and $|\overline{y_i}|$ is the estimated time series and $n$ is the number of non-missing data points.

## 3. Results

### 3.1. ERA5 Data Validation

The statistical analyses were based on the linear regression that compared the data of the weather stations versus the ERA5 product (Table 3). To facilitate the comparison, the tables were grouped by ecoregions, as shown in Table 1. In the Mixed Ecoregion the $R^2$ coefficient was higher for the Swift Current (0.75, $p < 0.001$), Rock Point (0.71, $p < 0.001$)), and Beechy stations (0.70, $p < 0.001$). The Taber station presented $R^2$ of 0.51 ($p < 0.001$) and, finally, the Coronach station had the weakest relationship, where $R^2$ was 0.23 ($p = 0.002$). In the Moist-Mixed Ecoregion the strongest $R^2$ relationships were Outlook (0.75, $p < 0.001$), Queenstown (0.67, $p < 0.001$), Buffalo (0.53, $p < 0.001$), and Scott (0.50, $p < 0.001$). The Last Mountain station obtained $R^2$ of 0.41 ($p < 0.001$), the weakest in this ecoregion. It is worth noting that the smallest RMSE were observed in the Mixed Ecoregion (Rock Point = 82.90 mm and Taber = 93.67 mm), while the highest RMSE values were in Coronach (188.51 mm) and Buffalo (203.85 mm), located in the Mixed and in the Moist-Mixed Ecoregions, respectively. The MBE results were like the RMSE, with the lowest values (greater proximity to in-situ observations) occurring in Taber (MBE= 62.81 mm) and Rock Point (MBE = 63.12 mm), both located in the Mixed Ecoregion. The highest MBE values were also in line with the RMSE results, especially for Coronach (MBE = 159.49 mm) and Buffalo (MBE = 188.96 mm). The MBE showed that the ERA5 overestimated the precipitation values in the ten analyzed locations.

**Table 3.** Linear regression between the observed data and ERA5-ECMWF (mm.yr-1) for the weather stations located in the Mixed Ecoregion and in the Moist = Mixed Ecoregion.

| Mixed Ecoregion | Equation | $R^2$ | Sig | RMSE | MBE |
|---|---|---|---|---|---|
| Beechy | y = 0.8931x + 150.88 | 0.71 | $p < 0.001$ | 122.63 | 111.43 |
| Coronach | y = 0.4107x + 359.21 | 0.23 | $p = 0.002$ | 188.51 | 159.49 |
| Rock Point | y = 0.7525x + 163.71 | 0.71 | $p < 0.001$ | 82.9 | 63.12 |
| Swift Current | y = 0.812x + 177.47 | 0.76 | $p < 0.001$ | 119.28 | 109.50 |
| Taber | y = 0.6888x + 179.95 | 0.52 | $p < 0.001$ | 93.67 | 62.81 |
| Moist-Mixed Ecoregion | Equation | $R^2$ | Sig | RMSE | MBE |
| Buffalo | y = 0.7299x + 273.08 | 0.53 | $p < 0.001$ | 203.85 | 188.96 |
| Last Mountain | y = 0.5643x + 274.82 | 0.42 | $p < 0.001$ | 140.43 | 111.65 |
| Outlook | y = 0.8719x + 158.3 | 0.76 | $p < 0.001$ | 122.67 | 113.71 |
| Queenstown | y = 0.7385x + 206.21 | 0.67 | $p < 0.001$ | 109.71 | 99.32 |
| Scott | y = 0.5959x + 252.58 | 0.51 | $p < 0.001$ | 123.59 | 107.99 |

### 3.2. Monthly Precipitation—Boxplots and the MAPE

The monthly precipitation boxplots from 1981 to 2019 are presented in Figures 2 and 3. In the Mixed Ecoregion it was found that the rainy months were May, June (the wettest of all), and July for both datasets (in-situ and ERA5). The months of April, August, and September were months of transition in terms of precipitation amount. Finally, the driest months were January, February, and December. The updated parameterizations of ERA5 were able to identify the outliers of precipitation in the study area in May and in transition months such as August and September.

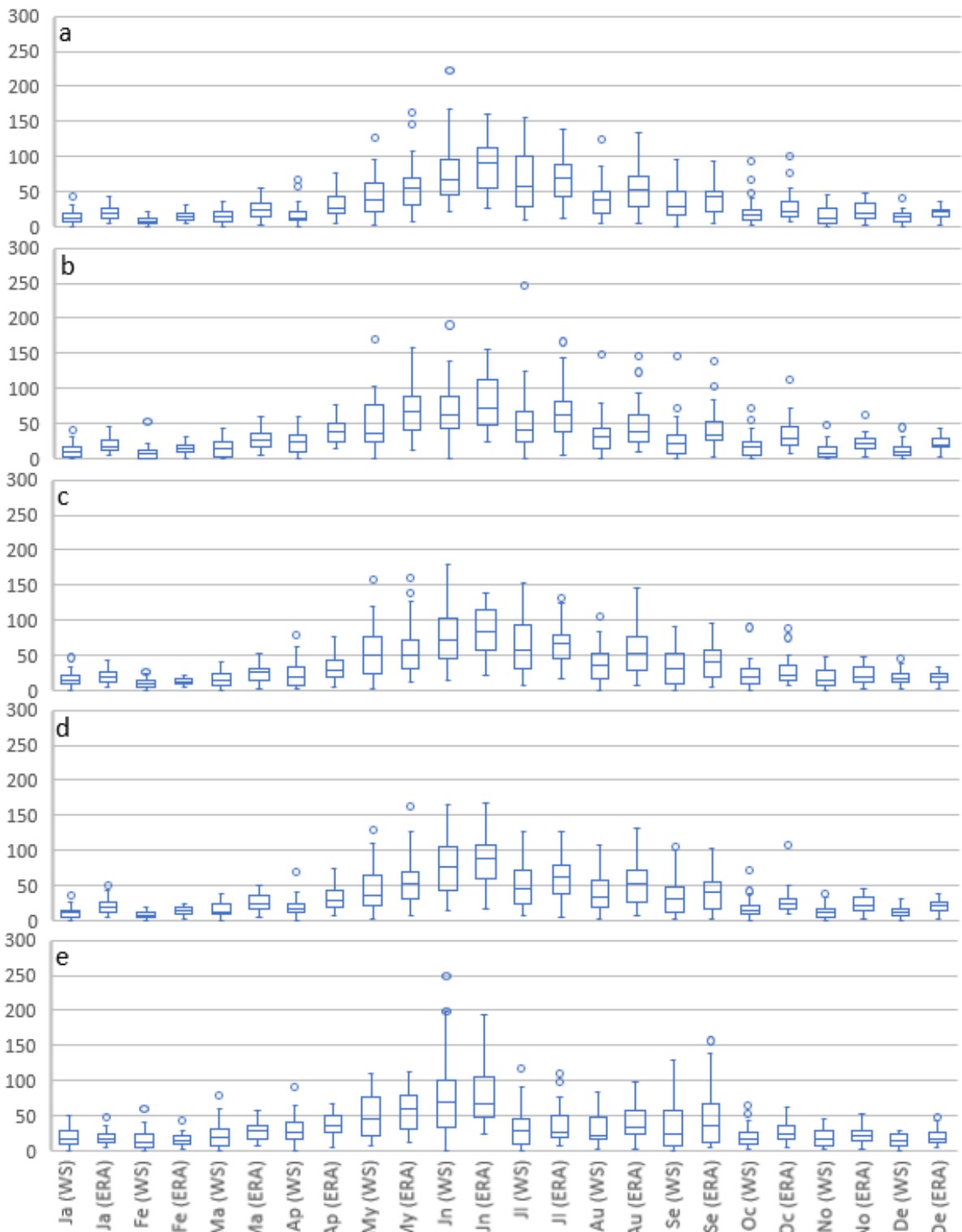

**Figure 2.** Boxplot of monthly precipitation (mm)—First the weather wtation (WS) data, followed by the ERA5-ECMWF (ERA) data (mm) in the Mixed Ecoregion for Beechy (**a**), Coronach (**b**), Rock Point (**c**), Swift Current (**d**), and Taber (**e**) in the period 1981–2019. The horizontal lines inside the boxes represent the median and the vertical lines at the top and bottom of the box represent the third and first quartiles, respectively. The ends of the vertical lines indicate the maximum (upper) and minimum (lower) values, and the isolated points show outliers.

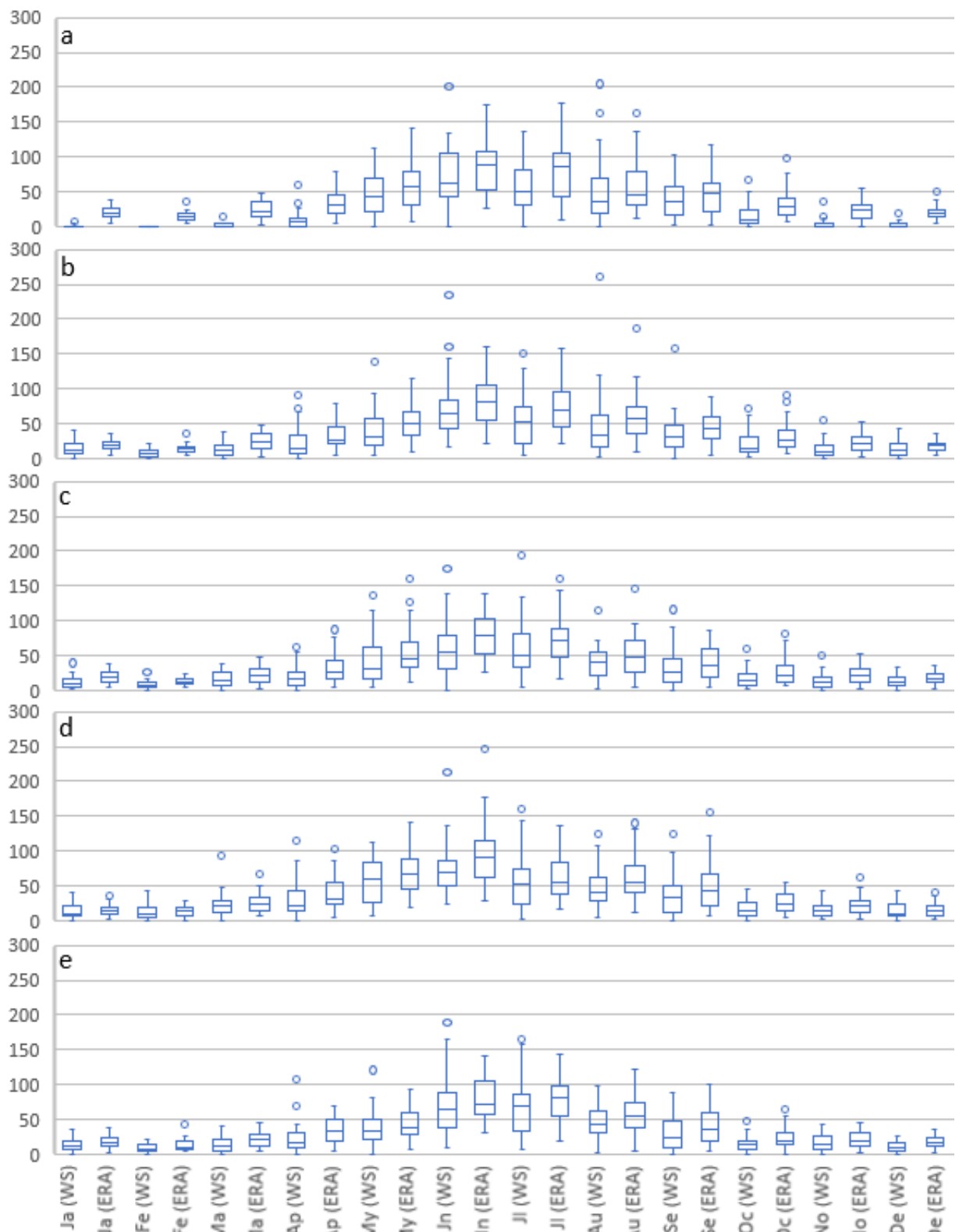

**Figure 3.** Boxplot of monthly precipitation (mm)—First the weather station (WS) data, followed by the ERA5-ECMWF (ERA) data (mm) in the Moist-Mixed Ecoregion for Buffalo (**a**), Last Mountain (**b**), Outlook (**c**), Queenstown (**d**), and Scott (**e**) in the period 1981–2019. The horizontal lines inside the boxes represent the median and the vertical lines at the top and bottom of the box represent the third and first quartiles, respectively. The ends of the vertical lines indicate the maximum (upper) and minimum (lower) values, and the isolated points show outliers.

We identified that in the Mixed Ecoregion the month of June concentrated the highest amounts of precipitation in both datasets (20.53% (WS) and 17.76% (ERA5) of the average annual precipitation of the five analyzed stations). However, the second wettest month showed divergences between the datasets. While the in-situ data indicated July (14.21% vs. 12.82%) as the second wettest month, the ERA5 data indicated the month of May (13.16% vs. 13.44% of the weather station). Regarding the dry months, there was agreement between both datasets. February was the month with the lowest precipitation with 3.00% (WS) and 3.23% (ERA5), followed by January with 4.09% (WS) and 4.33% (ERA5), and by December 4.15% (WS) and 4.39% (ERA5). In the Mixed Ecoregion the smallest differences between both datasets were observed during the dry period. In the Moist-Mixed Ecoregion most of the precipitation was concentrated in June, with 19.61% (WS) and 17.63% (ERA5) of the annual average precipitation in the five locations analyzed. Other months showed relevance during the wet period, such as July 16.58% (WS) and 15.40% (ERA5), August 13.11% (WS) and 12.07 (ERA5), and May 12.68% (WS) and 11.75% (ERA5). During the dry period the months that stood out the most were February 2.38% (WS) and 3.05% (ERA5), January 3.28% (WS) and 4.04% (ERA5), and December 3.31% (WS) and 3.87 (ERA5). Therefore, wet, and dry periods for both datasets were similar in terms of monthly records. The only divergence was the month of January which, from the in-situ data, was considered the second least rainy month and from the ERA5 data it was the third least rainy month.

The MAPE results help to identify where (which station) and when (which month) the two datasets had the greatest similarities and greatest divergences. In the Mixed Ecoregion (Figure 4) the station that presented the greatest similarities between the in-situ data was Rock Point, with the highest value of R (0.91) being observed in the month of May and the lowest value of MAPE (23.50) being observed in the month of June. On the other hand, the station that presented the biggest differences was Coronach. It is worth noting that, in general, in the Mixed Ecoregion, the month that presented the highest values of R and the lowest values of MAPE was May, while the month of February was the opposite.

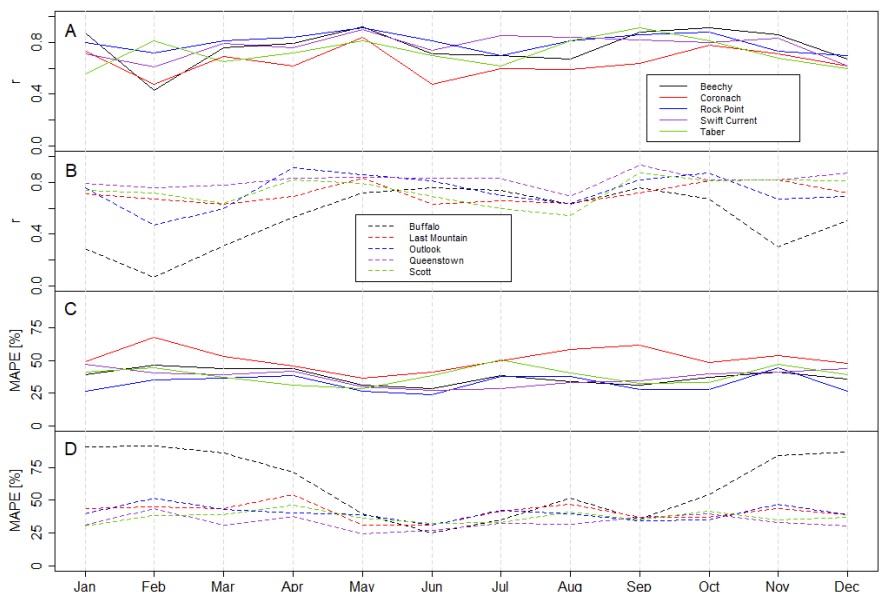

**Figure 4.** Monthly correlation coefficients and MAPE results for stations located in the Mixed Ecoregion and in the Moist-Mixed Ecoregion. Graphs (**A**,**B**) indicate, respectively, the monthly R values between measurements made by weather stations and ERA5 estimates in the Mixed Ecoregion and the Moist-Mixed Ecoregion. Graphs (**C**,**D**) indicate, respectively, the monthly MAPE values between measurements made by weather stations and ERA5 estimates in the Mixed Ecoregion and the Moist-Mixed Ecoregion.

In the Moist-Mixed Ecoregion (Figure 4) the greatest similarities were observed in Queenstown, especially in the month of December (R = 0.87) and May (MAPE = 24.70). The biggest differences were observed at the Buffalo station. It is also worth noting that excluding Coronach and Taber in the rest of the cases in the two ecoregions, the lowest MAPE values were observed in June.

### 3.3. Monthly Maps—ERA5

Figure 5 shows the spatial distribution of precipitation in the Mixed Ecoregion and in the Moist-Mixed Ecoregion in the period of 1981 to 2019. The highest precipitation records occurred in the central and southeastern region (SE) of the Mixed Ecoregion and in the eastern (E) and western (W) regions of the Moist-Mixed Ecoregion. On the other hand, December (Figure 5L), January (Figure 5A), and February (Figure 4B) were the driest months. It is also worth mentioning the transition months, such as May (Figure 5E) and August (Figure 5H). In May, greater precipitation was observed in the central and SE regions of the Mixed Ecoregion. In the Moist-Mixed Ecoregion, the wettest regions were E and W. When comparing both ecoregions in the months of May and July, an opposite spatial pattern is observed.

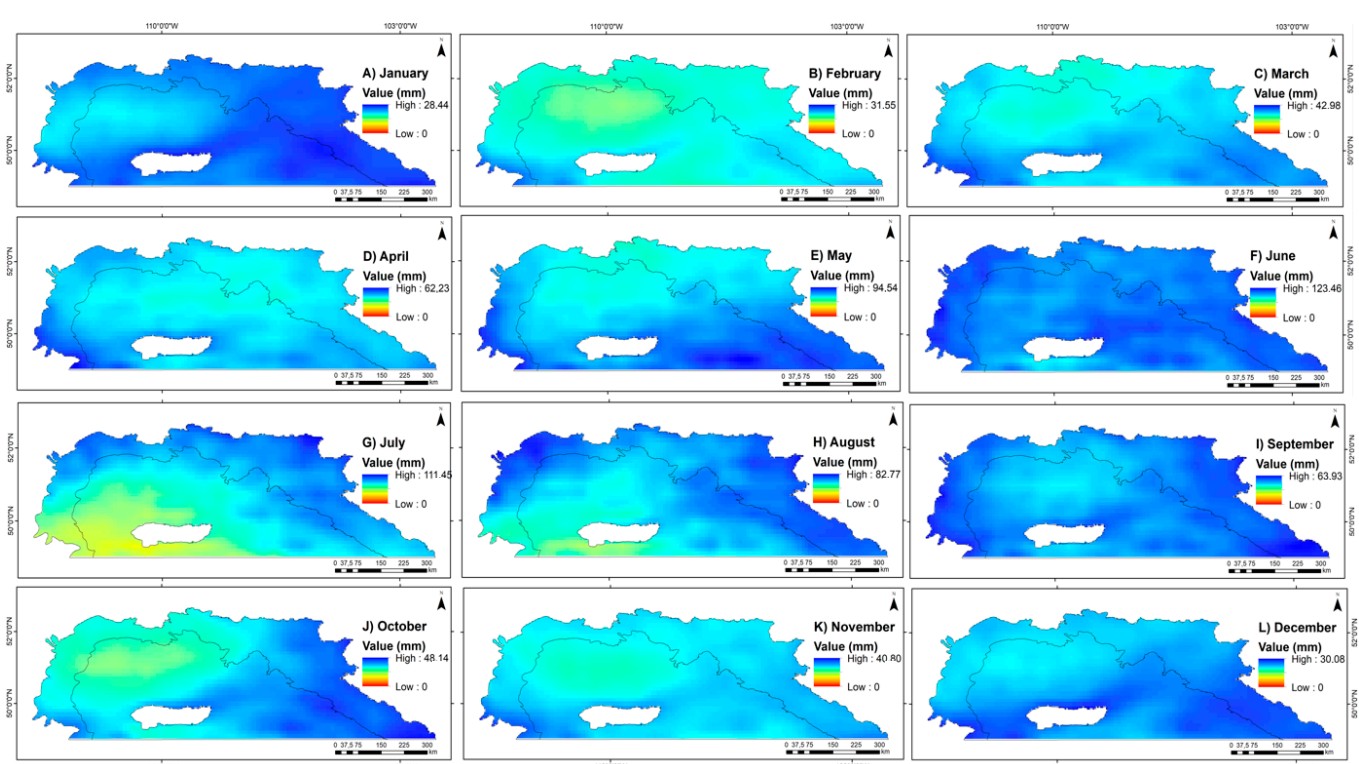

**Figure 5.** Average monthly precipitation—ERA5-ECMWF (mm monthly$^{-1}$) in the period 1981–2019.

### 3.4. SPI Derived from ERA5

In this section the SPI was derived from the ERA5 data for the detection of abnormal events, especially those related to historical El Niño-Southern Oscillation (ENSO) events. Figures 6 and 7 show the annual-SPI for the ten ERA5 pixels referring to the location of the weather stations for the period 1981–2019. Different colours describe wet, normal, and dry periods, according to the classification of the SPI (Table 2). For validation purposes, the SPI/ERA5 results were compared with years of the occurrence of the phases (El Niño and La Niña) in the climate variability mode of strong or very strong intensity. As a reference, the Oceanic Niño Index (ONI) was used, which helps to identify the El Niño events, responsible for causing a decrease in precipitation in the study area and La Niña events, which contribute to the opposite effect [46]. For comparison,

events were categorized as strong and very strong for El Niño and strong events for La Niña between 1981 and 2019, according to the ONI index. El Niño's strong years were 1987–1988 and 1991–1992 and the very strong years were 1982–1983, 1997–1998, and 2015–2016. The following years are associated with strong La Niña events: 1988–1989, 1998–1999, 1999–2000, 2007–2008, and 2010–2011.

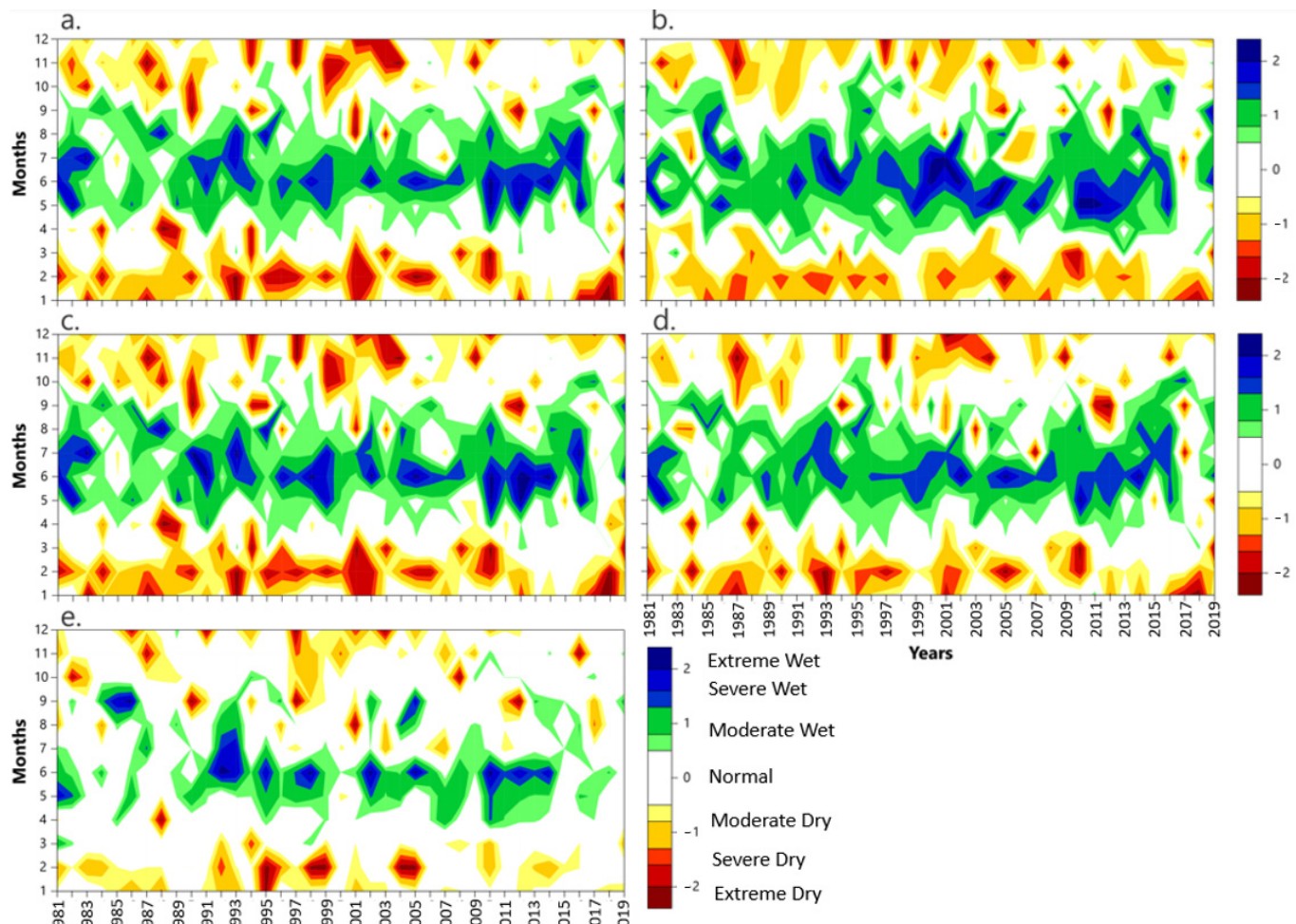

**Figure 6.** SPI derived from ERA5 data for the Mixed Ecoregion for Beechy (**a**), Coronach (**b**), Rock Point (**c**), Swift Current (**d**), and Taber (**e**) in the period 1981–2019.

In this study, the detection of an ENSO event was made by observing at least one month classified as moderate, severe, or extreme drought/wet. In the Mixed Ecoregion, no evidence was observed associated with La Niña in 1988–1989 in Taber, moderate events were detected (El Niño from 1982–1983 in Swift Current) and La Niña (1988–1989 in Coronach) and the others were classified as severe. During the 1990s, the 1991–1992 El Niño was classified as moderate in Beechy and the remaining ENSO events were classified as severe. Finally, in the 2000s the El Niño events (2015–2016) in Coronach and Rock Point, as well as the 2007–2008 La Niña in Taber were classified as moderate. The other events indicated by the ONI index in the 2000s were classified as severe.

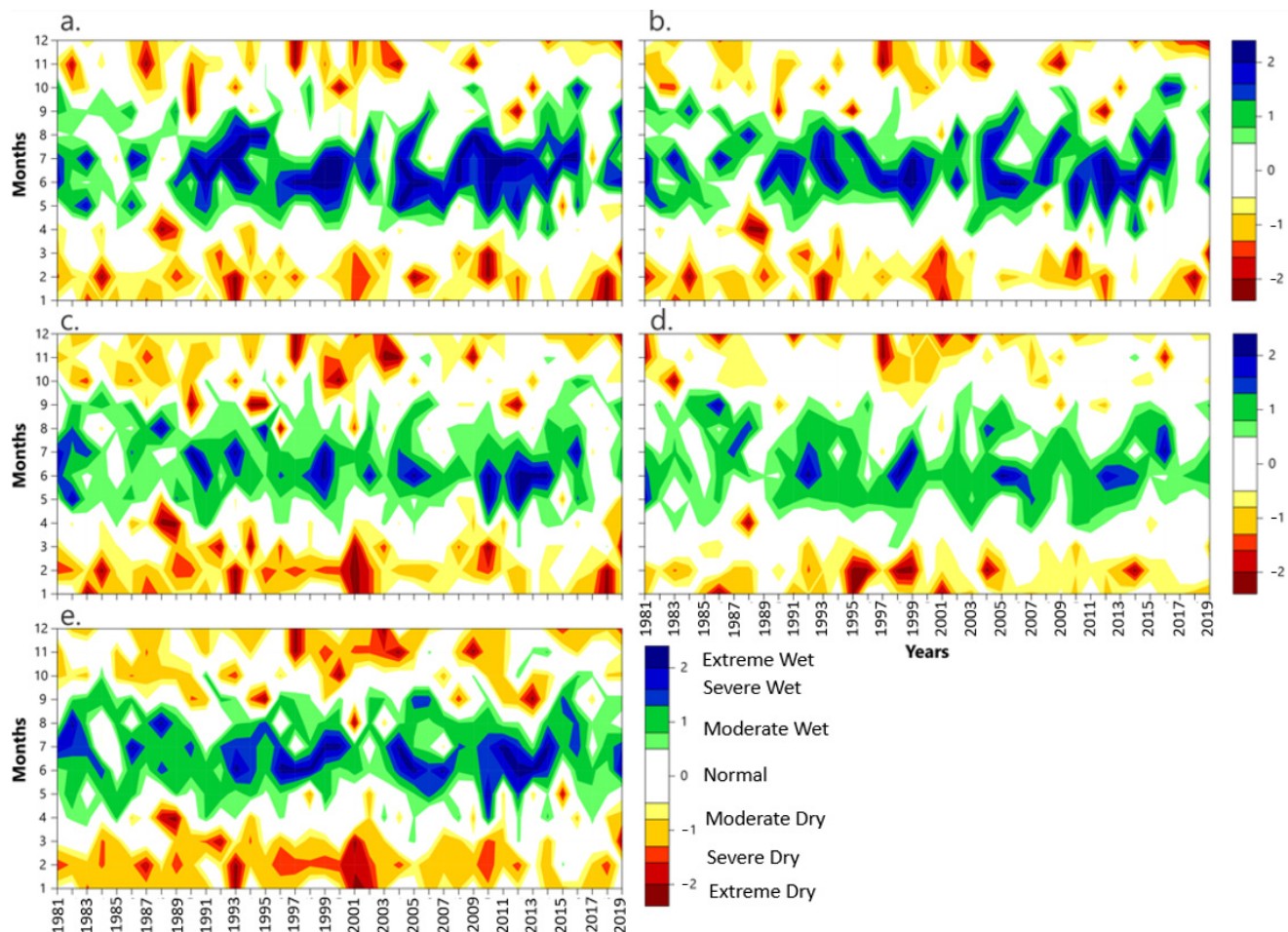

**Figure 7.** SPI derived from ERA5 data for the Moist-Mixed Ecoregion for Buffalo (**a**), Last Mountain (**b**), Outlook (**c**), Queenstown (**d**), and Scott (**e**) in the period 1981–2019.

In the Moist-Mixed Ecoregion, all El Niño events that occurred during the 1980s and that were highlighted in the ONI index were detected by the SPI derived from ERA5 data. However, different classifications of SPI were observed: in Scott (1982–1983) the classification indicated moderate droughts and in the other cases SPI indicated severe droughts. It is worth mentioning the months of January and February of 1987 in Last Mountain, Outlook, and Scott where unusual droughts were observed. As in the findings of [47–49], above normal precipitation was observed in the following summer after El Niño events. Regarding the La Niña events (1988–1989), severe wet events were observed in all locations. During the 1990s all ENSO events were detected by the SPI and classified as severe. In the 2000s, again all ENSO events were detected, but the 2015–2016 El Niño was classified as moderate in Last Mountain and Outlook, as were, respectively, the 2007–2008 and 2010–2011 La Niña in Outlook and Queenstown. The other detected events were classified as severe. Following the pattern suggested previously, wetter summers were observed shortly after El Niño events during the 1990s and 2000s.

## 4. Discussion

This study sought to demonstrate whether ERA5 has the potential to fill gaps or be used in regions of two ecoregions located in the Canadian Prairies with few meteorological stations or short time series. To quantify accuracy and precision between in-situ and ERA5 data, we used four metrics: $R^2$, RMSE, MBE, and MAPE. In this context, other studies carried out comparative analyses and obtained similar results ($0.25 > R^2 > 0.81$), such as [50] who performed the validation of the GPM versus in-situ

stations located in Southern Canada. However, the regression results were lower when compared to studies carried out in tropical regions of the planet. Studies like [16,51], obtained $R^2$ > 0.74, when validating the CHIRPS data versus data from in-situ stations in Brazil. Also in Brazil, [10] in an intercomparison study between data from the Global Precipitation Climatology Center (GPCC) and in-situ data obtained $R^2$ > 0.82. The studies carried out in Cyprus by [52,53] also obtained $R^2$ > 0.70, when comparing data from in-situ stations versus data from CHIRPS and GPM, respectively. Although our results, when compared to the results mentioned above, show lower coefficients of determination, it should be noted that the climate in the Canadian Prairies has specific characteristics, for example, the great distance from the coast, with no maritime effect in any of the ecoregions (Figure 1), followed by the low variation of the altimetric gradient (Figure 1 and Table 1), which implies precipitation events of low intensity [54–56] that normally contribute to the increase in deviation in estimated data. Therefore, physiographic factors have little influence on precipitation, being influenced by atmospheric circulation, particularly low-pressure systems that are responsible for advected humidity [51]. In this context, ERA5 was able to characterize the precipitation events in the transition months (April, May, September, and October) and with reasonable results for June and July in the study area (indicated by the MAPE values and by the monthly maps), which makes sense due to the precipitation existing in the spring is the result of atmospheric events of greater scale and that tend to cover a larger area, which would contribute to a greater similarity of the results, while summer precipitation often occurs as a result of the warming of the Earth's surface, having a more local character and, therefore, with greater spatial variability [57]. All these results combined show that ERA5 has the potential to fill gaps in time series or as a primary data source in regions with few weather stations.

Although the determination coefficients are not the highest observed in the literature, it must be considered that weather stations and ERA5 represent distinct forms of data acquisition and spatial integration of a phenomenon that has high spatial variability. It should also be considered that there may be biases in the data from weather stations, such as technical problems with the equipment and the human factor in conventional stations, especially in older observations, which affects the consistency of the time series and increases the error in relation to the observations of ERA5. For example, at the Buffalo weather station, the precipitation values collected during the winter, mainly in February, are extremely low compared to the ERA5 observations and to the observations made by the other weather stations used in this study for the same period, which suggests a local bias and not that the ERA5's accuracy was so low. In this context, local automated precipitation measurements are critical for regional scale rainfall and snowfall measurement and are often used for validation of precipitation rates estimated via satellites (TRMM, PERSIANN-CDR, among others) or climate modeling (GPCC, CRU, CHIRPS, among others). However, the precipitation measured at gauges is affected by undercatch, which is generally greater for precipitation in the solid form due to meteorological factors and/or flaws in their design to measure snowfall [15,58]. In the literature there are different "gauge-undercatch" correction factors, for example, the dynamic correction model (uses synoptic observation of variables, such as wind speed at the edge of the gauge, air temperature, relative humidity and intensity and phase of precipitation) and the fixed climatology approach [59], and the choice of correction factor will impact the estimation of multiscale precipitation, mainly at high latitudes [60]. It is well known that gridded precipitation information is designed to produce a full coverage product [61]. However, when comparing ERA5 data with the Global Historical Climate Network (GHCN), both distance from shore and elevation difference affect estimates in the USA, as well as in Canada [61]. ERA5 has less precipitation along the coast than the GHCN observations and greater amounts observed inland. These results and information reinforce the importance of data sources that are less susceptible to possible variations in the quality of the observed data. Even so, it was possible to observe

similar patterns and significant correlations between the two datasets, which makes the ERA5 interesting for applications that require precipitation monitoring at a specific time of year, such as forage insurance projects that are focused on the April through July period, such as the Forage Rainfall Insurance Program in Saskatchewan.

We used boxplots (Figures 2 and 3) to compare monthly averages derived from observations made by weather stations and ERA5. In general, it was found that ERA5 had higher total monthly precipitation, which could be related to its horizontal resolution (grid spacing) due to the greater information capture power of a grid cell compared to the punctual information captured by the weather station. Even though the horizontal resolution was improved and with the changes in precipitation and convection parameterization schemes when compared to the ERA-Interim, it is important to use the results with caution, since precipitation varies greatly spatially and the results can contain deviations [24]. Previously, the precipitation scheme was based on [62], being updated with representation of Mixed-phase clouds [63], and prognostic variables for precipitating rain and snow [64,65], while the convection scheme was based on [66], with the large-scale entrainment and coupling process being updated based on the large redistribution of precipitation from the Hadley cell to the Walker cell and, mainly, the diurnal cycle [67,68]. The results of June as the wettest month for both Mixed Ecoregion and Moist-Mixed Ecoregion corroborate the results obtained previously by [69,70], in which they indicated that the Southern region of the Canadian Prairies is characterized by greater records of precipitation in June due to the late-spring/early-summer position of the polar jet stream. In addition, the increase in humidity in Canadian Prairies is due to the humid air brought in from the Southern United States and the Gulf of Mexico [57]. Studies like [57,70], even without using gridded precipitation data, obtained similar results, with a greater record of precipitation between May and July in other regions of the Canadian Prairies. Therefore, the ERA5 data, in addition to presenting results significantly correlated with in-situ data at some times of the year, showed some consistency with results obtained in other studies in terms of identifying the months with the highest amounts of precipitation.

Monthly maps are important to understand the spatial pattern of precipitation in the two studied ecoregions. For example, we found that June (Figure 5F) stands out for being the rainiest month in the two ecoregions, a similar result to those obtained by [57,70,71]. We also found the regions that registered the lowest precipitation in May showed the highest records in July, which is important for ecological dynamics, such as the period of vegetation growth. This type of information is also relevant since both ecoregions are considered arid and are agricultural production areas, so it is crucial to rural producers to identify wetter periods [72]. Ref [57] state that although the growing season of 1954 has one of the highest records of precipitation amount, the crop production was approximately 30% below normal due to the lack of precipitation at the beginning of the growing season. As a result, the production cycle was affected due to delays in the planting, maturation, and harvesting of crops.

The Canadian Prairies are known to be a drought-susceptible region, particularly in southwestern Saskatchewan and southeastern Alberta [34]. In this context, the graphs derived from SPI/ERA5 help to identify extreme events during the period 1981–2019. In their study [57] concluded that droughts are caused by a stronger ridge which blocks the flow of moisture-laden air from the west. Some drought events were so remarkable, such as the period from 1999 to 2005, that they required specific characterization studies [72,73]. According to the literature, the period 1999–2005, 2001 drew attention due to the expansion of the drought during the summer, with a peak in the winter (including the beginning of 2002). Based on the results of this study, it is possible to affirm that the SPI was able to detect several abnormal drought events, including the period 1999–2005. Moreover, based on the analysis of the SPI/ERA5 graphs, it was possible to conclude that the ERA5 data were sensitive to drought events, as observed in Buffalo, Outlook, and Scott. One of the hypotheses that can contribute to

the extreme event detection capability of ERA5 data is the fact that a pixel integrates information from a larger area than the area covered by a weather station. These results are in line with the studies by [70,74].

## 5. Conclusions

This study performed multiple comparisons to identify whether ERA5 data is a reliable source of precipitation information in cases of absence of weather stations or gaps in the historical series. The study area is far from the coast and the topography is generally flat, factors that contribute to the precipitation being highly dependent on atmospheric circulation and high temperatures, which results in convective precipitation in the summer months, as evidenced by data from the weather stations and ERA5.

The results of comparisons showed that ERA5 has the potential to be a source of data in the Canadian Prairies in cases of gaps in the time series or few meteorological stations in the studied area, since nine cases of significant relationship (with the $R^2$ varying between 0.42 and 0.76) at the 99% probability level ($p < 0.01$) between ERA5 and weather stations data, as well as the results presented by MAPE in which the month of May had the biggest agreement between both datasets and the month of February with the biggest divergences in the two studied ecoregions. However, it is noteworthy that, even with the update of the precipitation and convection parameterizations, the ERA5 data overestimated the monthly records in the two ecoregions analyzed, which can be a limiting factor for studies that require greater accuracy of the results, such as precision agriculture. The SPI/ERA5 data was able to detect wet periods and highlight dry periods due to the spatial coverage of ERA5 product. Also, a general agreement between SPI/ERA5 and the ENSO events during the period analyzed was found.

**Author Contributions:** Conceptualization, T.F. and C.A.d.S.J.; methodology, T.F., C.A.d.S.J., P.E.T. and J.F.d.O.-J.; validation, T.F., C.A.d.S.J., K.J.C., P.E.T. and J.F.d.O.-J.; formal analysis, T.F. and X.G.; investigation, T.F.; data curation, T.F. and K.J.C.; writing—original draft preparation, T.F.; writing—review and editing, C.A.d.S.J., K.J.C., P.E.T., J.F.d.O.-J. and X.G.; visualization, T.F. and K.J.C.; supervision, T.F. and X.G. All authors have read and agreed to the published version of the manuscript.

**Funding:** This research was funded by the University of Saskatchewan and the Natural Sciences and Engineering Research Council of Canada (NSERC).

**Data Availability Statement:** The datasets used and/or analyzed during the current study are available from the corresponding author on reasonable request.

**Conflicts of Interest:** The authors declare no conflict of interest.

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
