# Peer review of "Is the Gridded Data Accurate? Evaluation of Precipitation and Historical Wet and Dry Periods from ERA5 Data for Canadian Prairies"

_remotesensing, doi:10.3390/rs14246347_

Round 1

Reviewer 1 Report (New Reviewer)

The author described the significance of this article very clearly. I have one major concern regarding the spatial matching of station data to gridded ERA5 data. The spatial resolution of ERA5 precipitation data is 9 km, and precipitation is likely to show spatial heterogeneity within 9 km × 9km. The author directly selected ERA5 pixels that cover the weather stations, and I think this treatment may be problematic. The specific comments are as follows.

Line 138: Please add latitude and longitude labels in Figure 1.

Line 167: Please rewrite this sentence.

Line 173: 38-year or 39-year?

Line 180: α in Equation 1 is not clear.

Line 230: mm.yr-1?

Author Response

Dear reviewer,

Thank you for your comments and suggestions. Responses to your contributions are in the attached file.

Reviewer 2 Report (New Reviewer)

Line 30: Make sure to define the acronyms (MAPE) the very first time that they are used in the manuscript. 

Line 43-80: One of the main limitations of the satellite products is their poor performance over snow and ice surfaces which is the key feature of your study area. Please mention that in this paragraph to clarify the situation for the readers.

Line 72: You have used "almost global" frequently in the text, you may replace it with "quasi-global" as this is the common terminology used in the community. 

Gauge observations suffer from gauge-undercatch issues due to many factors such as wind. This is a more serious issue in higher latitudes and colder climates. Please explain what would be the impact of this phenomenon on your results.

Line 91-93: This is not necessarily a true statement, there are many studies showing that ERA-Interim has the same skill as ERA5, though the spatiotemporal resolutions offered by ERA5 are indeed better.

Figure 1: Add lon-lat ticks to the maps.

Table 1: It can be moved to supplementary materials. 

Section 2.5: Rename it to statistical metrics. Also, it can be presented as an appendix. 

Figures 2 and 3: You are showing only one set of boxplots (blue), which is for ERA5 or stations? Very confusing! Clarify the plot. If that's the difference between ERA5 and stations, it is not clearly communicated.

Tables 4 and 5 can be replaced by a simple line plot. No need to include all details here. 

Figure 4: The quality of the plot should really improve, I would recommend using programming languages to create quality figures instead of whatever GIS application is used to create individual plots and then paste them together. The colorbar doesn't seem to be very consistent with the colors in the maps!

Figures 5 and 6: Fonts are very small and hard to read. 

Author Response

Dear reviewer,

Thank you for your comments and suggestions. Responses to your contributions are in the attached file.

Reviewer 3 Report (New Reviewer)

The article entitled " Is the gridded data accurate? Evaluation of precipitation and historical wet and dry periods from ERA5 data for Canadian Prairies" evaluate the state of art ERA5 data from ECMWF against ten (10) in-situ observation using four metrics (i.e., R2. RMSE, MBE and MAPE) in two ecoregions of the Canadian Prairies from 1981-2019. Also, the intraseasonal variability, SPI-based drought assessment and its relationship with ENSO were conducted. Quantitative results obtained showed fairly close agreement and statistical significance between in-situ observation and ERA5. The SPI/ERA5 data showed anomalies in temporal patterns are consistent with literature.

Overall, the manuscript is well written and organized. I find the results section very interesting and provides useful information for the readers. The results are complemented with significant figures to help in results visualization.

However, I have few suggestions for the authors to improve their manuscript in the abstract and conclusion sections. Also, I noticed few typos and unclear statements that need correction and clarification, respectively. I recommend moderate language editing to improve the flow.

The detailed comments are provided as track changes in the pdf document attached to the authors.

Author Response

Dear reviewer,

Thank you for your comments and suggestions. The responses to your contributions are in the attached file, more precisely within the boxes indicated by the yellow underlines.

Round 2

Reviewer 1 Report (New Reviewer)

I appreciate your efforts to improve the manuscript as recommended. I accept the current manuscript form for publication in the Remote Sensing journal.

Author Response

Dear Reviewer,

The authors are grateful for your suggestions and comments, they contributed to the manuscript to be improved.

Reviewer 2 Report (New Reviewer)

The authors have addressed most of my questions. However, regarding the gauge-undercatch and also issues of PMW over cold regions (~ high latitude) and over frozen surfaces, you need to cite better references. There are many good articles published which are specifically focused on these topics. I recommend a minor revision, the work can be accepted after addressing this problem. 

Author Response

Dear Reviewer,

The authors are grateful for your suggestions and comments, they contributed to the manuscript to be improved.

Attached is the document with the information added from your comments.

This manuscript is a resubmission of an earlier submission. The following is a list of the peer review reports and author responses from that submission.

Round 1

Reviewer 1 Report

Please see attached review document.

Reviewer 2 Report

The article is devoted to the actual topic of studying the quality of estimation of the precipitation amounts over land according to the ERA5 reanalysis data at different space-time scales. Two ecoregions in the Canadian Prairies were chosen as the area of ​​interest, which are relatively poorly covered by both weather station observations and satellite measurements.

It is shown that, at monthly averaging intervals, the reanalysis data generally agree satisfactorily with the data of the selected weather stations (although there are cases of a rather low correlation). At the same time, in general, the estimates based on the reanalysis data slightly exceed the results of observations at weather stations. It is concluded that reanalysis data are generally suitable for studying phenomena caused by large-scale atmospheric circulation processes, however, in detailed studies, reanalysis data can be used with some caution. This is in general agreement with other known works.

There are the following remarks to the article.

1. Typos in equation (3) that need to be corrected.

2. Legends in fig. 2 – 5 are not detailed enough (what does the gray stripe in fig. 2, 3 mean; what do boxes, bars and diamonds of different colors correspond to in fig. 4, 5?)

3. Captures on the lower axes of fig. 4 and 5 are hard to read.

I recommend the work for publication after the elimination of these shortcomings.

Reviewer 3 Report

The manuscript “Is the gridded data accurate? Evaluation of precipitation and historical wet and dry periods from ERA5 data for Canadian Prairies” evaluates ERA5's ability to model the areal and temporal distribution of precipitation for selected areas of Canada. The study is very comprehensive and attempts to answer several questions. Unfortunately, the authors use only 10 stations to compare model results, arguing that only these stations have complete data for the period 1981 to 2019. I believe this number is too small to produce robust results.

Maybe I missed something, but it's not clear from the text how specifically the grid data was compared to the station data. Was the value from the nearest grid compared? What is the effect of the area corresponding to grid values on the comparison with the point measurement value?

The text compares the relative amounts of precipitation in each month. It would be very helpful if these values were expressed in terms of their precision/uncertainty.

Specific comments:

L25: Please add “daily means” (data measured from 1981-2019).

L159: What do you mean by “We calculated the α and β parameters of gamma for each (ERA5 data) on the annual scale.“ You calculate month characteristics but distribution parameters are calculated from annual averages? Please, explain you data processing.

L197: Please, be more specific “are lower“.

L209: Can you add some arguments why „were acceptable“?

L226: Please explain in details “which could be related to its horizontal resolution”.

Reviewer 4 Report

From my point of view, this is an interesting study that can have a positive impact for future studies regarding the analysis and reconstruction of meteorological phenomena in regions with a few stations or gaps in the record.

Round 2

Reviewer 1 Report

Please see attached review document.
